# Can Quality Improvement Methodologies Derived from Manufacturing Industry Improve Care in Cardiac Surgery? A Systematic Review

**DOI:** 10.3390/jcm11185350

**Published:** 2022-09-12

**Authors:** Paulien Christine Hoefsmit, Stijn Schretlen, George Burchell, Jaap van den Heuvel, Jaap Bonjer, Max Dahele, Reinier Zandbergen

**Affiliations:** 1Department of Cardiothoracic Surgery, Amsterdam University Medical Centre, 1081 HV Amsterdam, The Netherlands; 2Integrated Health Solutions, Medtronic Inc., 5616 VB Eindhoven, The Netherlands; 3Medical Library, Vrije Universiteit, 1081 HV Amsterdam, The Netherlands; 4Department of Healthcare Management, University of Amsterdam Business School, 1018 TV Amsterdam, The Netherlands; 5Department of Surgery, Amsterdam University Medical Centre, 1081 HV Amsterdam, The Netherlands; 6Department of Radiation Oncology, Amsterdam University Medical Centre, 1081 HV Amsterdam, The Netherlands

**Keywords:** Quality Improvement Methodologies, cardiac surgery, Lean, Six Sigma, Toyota Production System

## Abstract

Objectives: Healthcare is required to be effectively organised to ensure that growing, aging and medically more complex populations have timely access to high-quality, affordable care. Cardiac surgery is no exception to this, especially due to the competition for and demand on hospital resources, such as operating rooms and intensive care capacity. This is challenged more since the COVID-19 pandemic led to postponed care and prolonged waiting lists. In other sectors, Quality Improvement Methodologies (QIM) derived from the manufacturing industry have proven effective in enabling more efficient utilisation of existing capacity and resources and in improving the quality of care. We performed a systematic review to evaluate the ability of such QIM to improve care in cardiac surgery. Methods: A literature search was performed in PubMed, Embase, Clarivate Analytics/Web of Science Core Collection and Wiley/the Cochrane Library according to the Preferred Reporting Items for Systematic Reviews and Meta-Analysis methodology. Results: Ten articles were identified. The following QIM were used: Lean, Toyota Production System, Six Sigma, Lean Six Sigma, Root Cause Analysis, Kaizen and Plan-Do-Study-Act. All reported one or more relevant improvements in patient-related (e.g., infection rates, ventilation time, mortality, adverse events, glycaemic control) and process-related outcomes (e.g., shorter waiting times, shorter transfer time and productivity). Elements to enhance the success included: multidisciplinary team engagement, a patient-oriented, data-driven approach, a sense of urgency and a focus on sustainability. Conclusions: In all ten papers describing the application of QIM initiatives to cardiac surgery, positive results, of varying magnitude, were reported. While the consistency of the available data is encouraging, the limited quantity and heterogenous quality of the evidence base highlights that more rigorous evaluation, including how best to employ manufacturing industry-derived QIM in cardiac surgery is warranted.

## 1. Introduction

Quality Improvement Methodologies (QIM) from the industry have been applied to healthcare to improve value, enhance access, control increasing expenditure and improve quality [1,2,3,4,5]. This has led to a number of publications concerning the application of such QIM to surgical care [6,7,8,9,10,11,12]. Lean methodology and Six Sigma are examples of QIM that have successfully improved performance in healthcare [13,14]. The fundamental approach to these QIM involves identifying the care pathway to be improved, defining the correct outcome metrics, and performing a baseline measurement, followed by process analysis, identifying inefficiencies and subsequently implementing improvements in the care pathway. Finally, measurements are performed to quantify whether the desired gains and improvements have been realised. In this paper, we focus on cardiac surgery, which is associated with an intensive and complex perioperative process, a large amount of supporting equipment, involvement of multiple well-trained multidisciplinary healthcare professionals and considerable expense [15]. Examples are the use of cardiopulmonary bypass, post-operative stay in the Intensive Care Unit (ICU) and intensive monitoring in the cardiac care unit or surgical ward (telemetry, vital signs, bloodwork, X-rays, electrocardiography) to prevent and manage post-operative complications, including cerebrovascular ischemic events, myocardial infarction, renal dysfunction, atrial fibrillation, cardiac tamponade and deep sternal wound infections [16]. Such complications can increase the length of in-hospital stay after cardiac surgery. Since cardiac surgery considerably impacts limited hospital resources [17,18,19,20,21], it is vital to maximise its quality and value by optimising care pathway efficiency and patient flow. Any additional stress on the healthcare system, such as that experienced during the COVID-19 pandemic, which reduces ICU and operating room capacity, only increases the necessity for efficiency and value [22,23,24]. Furthermore, inefficiencies are expensive and cause delays and cancellations, which may be followed by compromised patient outcomes and dissatisfaction [25,26]. As an example of the potential for cost savings, Hawkes et al. calculated that a better flow of patients through the care pathway in cardiothoracic surgery could save 45 million (M) pounds (52 M euros) annually in England [27]. This demonstrates that there is a considerable opportunity and potential for quality and value improvement in cardiac surgery [28]. We performed a systematic review to evaluate the effectiveness of QIM on the pre-, intra-, and postoperative care process in cardiac surgery. The focus of this review is specifically on manufacturing industry-derived QIM, one feature of which is a simultaneous focus on both quality improvement and process efficiency. It is not a review of other improvement measures that have had a major impact on quality and have previously been extensively reviewed in the literature, including, for example: large national or regional databases; use of risk-adjusted mortality rates; site visits; and advocating a minimum volume of procedures per unit, or per surgeon. 

## 2. Materials and Methods

A literature search was performed based on the methodology described in the Preferred Reporting Items for Systematic Reviews and Meta-Analysis (PRISMA) statement [29].

### 2.1. Search Strategy 

To identify relevant publications, we conducted systematic searches focused on the pre-, intra- and post-operative process in cardiac surgery and QIM in the bibliographic databases PubMed, Embase, Clarivate Analytics/Web of Science Core Collection and Wiley/the Cochrane Library from inception up to 15 February 2021. The search included keywords and free text terms for (synonyms of) ‘cardiac surgery’ combined with (synonyms of) ‘Quality Improvement Methodologies’. The particular QIM included in this paper are consistent with prior publications and literature [2,11]. A full overview of the search terms per database can be found in the Appendix A (see Appendix A). No systematic review protocol was registered. No limitations on date or language were applied in the search. Duplicate articles were removed. The references of the identified articles were searched for relevant publications. 

### 2.2. Selection Process 

Studies were included in the final review if they were published in a peer-reviewed journal, described the application of one of the QIM included in the search, were directed at improving the pre, intra- or post-operative process of adult cardiac surgery, written in English, available as a full article and published after 2000 (therefore in the last two decades). Only including English articles is in accordance with the strategy used by Moher et al. [30]. Studies were excluded if the article was a conference abstract or editorial. The cardiac surgery care pathway comprised the pre-, intra- and post-operative processes and was defined as the period between referral until discharge after cardiac surgery. Two reviewers (P.C.H. and R.Z.) independently screened potentially relevant articles and abstracts for eligibility. If necessary, the full-text article was assessed. Overall, there was good concordance between the reviewers, with minor differences resolved through consensus.

### 2.3. Data Assessment 

The full text of version articles was obtained. Two reviewers (P.C.H. and R.Z.) independently evaluated the full-text papers, and extracted relevant information, including author, year, country, duration, objective, number of analysed procedures, QIM, interventions applied and results. 

### 2.4. Data Analysis 

In this instance, consistent with the number of identified articles, we performed a review and summarised the details and outcomes of the included studies.

### 2.5. Assessment of Risk of Bias in Included Studies

Quality assessments of each included study were conducted using the Newcastle-Ottawa Scale for observational studies [31]. Three elements were scored: selection, comparability and outcome. The quality of the included studies was rated good, fair or poor. 

### 2.6. Quality Improvement Methodologies

The Toyota Production System is a complete quality improvement system that aims at the complete elimination of waste and the continuous pursuit of the most efficient processes [14]. Lean is a philosophy, of which the Toyota Production System is an example, that consists of five principles (value, value stream, flow, pull and perfection) and methods that create optimal value for patients and organisations by reducing waste, optimising resource utility, and improving efficiency on a continuous basis [13,14]. Tools used in Lean include Kaizen, process mapping, value stream mapping, and poka-yoke. Six Sigma is a more data-driven team-elaborated QIM focused on removing errors. Lean and Six Sigma are often combined to strive for operational excellence [13]. The approach of Define, Measure, Analyse, Improve and Control (DMAIC) is often used in Lean Six Sigma. Plan-Do-Study-Act (PDSA)/Plan-Do-Check-Act (PDCA) cycles are comparable methods and frequently applied to smaller-scale process improvement. Both are considered independent methods and a part of Lean. Statistical Process Control, also integrated into Six Sigma, monitors and controls a process by using tools such as run and control charts [32]. Total Quality Management engages all employees in continuous, customer-focused improvement to achieve excellence and success. It can be traced back to the early days of modern quality improvement and inspired Lean and Six Sigma. Clinical Audit is a method that checks if defined quality standards are met [2].

## 3. Results

The results of the search are summarised in Figure 1 and Table 1. After removing duplicates, the literature search generated a total of 1214 references: 251 in PubMed, 649 in Embase, 314 in Clarivate Analytics/Web of Science Core Collection and 3 in Wiley/the Cochrane Library. Reviewing the abstracts led to exclusion of 1179 articles. The main reasons for exclusion were unrelated to the pre-, intra- and post-operative cardiac surgery process, a non-manufacturing industry-derived QIM, or the reference was not available in English. After examination of the full text of the remaining 35 articles, a total of 10 articles met the inclusion criteria.

### 3.1. Descriptive Synthesis of the Results 

Seven of the ten articles originated from the United States of America (USA) and the remaining three from Canada, Sweden and the Netherlands. Most studies were published as quality improvement reports with a pre-vs. post-intervention study design. The various QIM could be divided into (1) larger-scale quality improvement systems: Lean (*n* = 3), Toyota Production System (*n* = 1), Six Sigma (*n* = 1) and the combination of Lean and Six Sigma (*n* = 1), or (2) individual quality improvement related-activities, including: PDSA (*n* = 4), root cause analysis (*n* = 5), process mapping (*n* = 3), value stream mapping (*n* = 2), pull methodology (*n* = 1) and Kaizen (*n* = 1). Berry et al. applied their self-developed ‘ProvenCare’ improvement programme, which included PDSA and resulted in the implementation of 40 process elements based on recommendations of the American College of Cardiology American Heart Association (ACC/AHA) guideline for CABG surgery [40]. Three articles redesigned the complete pre-, intra- and post-operative phases. The main, discrete improvements that were introduced and implemented as a result of applying the QIM can be summarised as: process elements and better teamwork, education and communication. Important process elements included the redesign of the transfer process, fast-tracking, the standardisation of perioperative drug administration, hygienic measures, glucose management, and implementation of checklists and extubation guidelines. Team training, education, stand-up meetings, pre-operative briefings, collaborative bedside rounds, work agreements and methods to improve communication were also impactful. More detailed information is included in Table 1. All articles showed improvements in both patient (infection rates, ventilation time, mortality, adverse events and glycaemic control) and process-related outcomes (shorter wait times, shorter transfer time and increased productivity), however, not all were statistically analysed or significant. Furthermore, improvements in patient and staff satisfaction and financial performance were reported.

### 3.2. Patient Related Outcomes 

The application of Six Sigma in a 350-bed regional medical centre resulted in reduced rates of surgical site infections from 3.74 to 0.7 per 100 procedures, and ultimately to 0 for 30 months, representing 590 procedures [35]. In-depth root cause analysis resulted in a decrease in the incidence rate of deep sternal wound infection from 5.1% to 0.9% during a 9-month period [37]. Compliance with infection control measures during the care of cardiac surgery patients was improved through the application of PDSA cycles [39]. Lower rates of mortality and major adverse events after CABG surgery following the application of Toyota Production System-based improvements were reported compared to the regional Society of Thoracic Surgeon database [34]. The ProvenCare programme decreased the number of complications from *n* = 53 (39%) to *n* = 41 (35%, *p* = 0.55) in a total of 254 CABG procedures over a period of 17 months [40]. Multiple outcomes were analysed, but only discharge location to home showed a significant improvement from 81% (*n* = 111) vs. 90.6% (*n* = 106, *p* = 0.03) [40]. A summary of the main interventions of the ProvenCare programme is provided in Table 1. The application of Lean methodology increased the proportion of patients extubated within 6 h from 27% to 50% (*p* = 0.0001) and decreased the median length of intubation from 9.7 to 6.1 h (*p* = 0.0019) [36]. Significant reductions in ventilation time were also reported by Hefner et al. [41]. Lean management led to shorter duration of ventilation from 11.4 h to 6.9 h (*p* < 0.001), fewer patients reintubated (11.8% to 3.5%, *p* = 0.08) and a lower rate of prolonged ventilation (29.4% to 8.6%, *p* = 0.004). Lean Six Sigma improved glycaemic control assessed using multiple parameters in the cardiac surgery ICU and there were significantly fewer hypoglycaemic events [42].

### 3.3. Process-Related Outcomes 

Geoffrion et al. redesigned the handoff process, resulting in a significantly shorter time to transfer from the operating room to the ICU (12.6 to 10.7 min) [33]. The ProvenCare programme focused on increasing the rate of performance of 40 discrete elements in patient care. The performance rate of all 40 elements, which can be found in Table 1, rose from 59% to 100% after the programme (*p* = 0.001) [40].

Watling et al. in Canada applied Lean as QIM during a period of 24 months, which resulted in 17 days shorter waiting time for cardiac surgery and increased annual production from 788 to 873 cardiac surgeries (10.8%), despite a 7.5% increase in cancellations, mainly due to limited ICU resources [38].

### 3.4. Patient and Staff Reported Outcomes

Geoffrion et al. reported improved staff satisfaction at 6 months (*p* < 0.02) and 2.5 years (*p* = 0.133) after applying a twelve-step implementation process, PDSA cycles and multiple rapid-cycle process improvements for redesigning the handoff process [33]. Culig et al. reported a constant patient satisfaction rate at the ICU of 99% and improved staff satisfaction after applying the Toyota Production System to the care pathway [34].

### 3.5. Financial Performance

Costs savings of US $3497 per CABG and US $884,000 in total were reported after the application of the Toyota Production System [34]. Kles et al. found that a reduction in surgical site infection rate led to estimated cost savings of US$606,498 in 590 CABG procedures over 30 months [35].

### 3.6. Risk of Bias Assessment

The quality of the included papers was assessed with the Newcastle-Ottawa Scale and was rated as following: good *n* = 4 [33,36,37,40], fair *n* = 1 [34] and poor *n* = 5 [35,38,39,41,42] (Table 2). 

### 3.7. Relevant Disclosures and Conflicts of Interests 

Relevant disclosures and conflicts of interest were reported for two studies (Table 3) [34,38]. The project of Culig et al. was supported by a grant. Waitling et al. had a partnership with Integrated Health Solutions, Medtronic.

## 4. Discussion 

We identified ten articles that used manufacturing industry-derived QIM to improve the pre-, intra- or post-operative cardiac surgery process. Seven out of the ten projects were performed in the USA. Although seemingly limited, the number of studies meeting the inclusion criteria is reasonable when compared to a similar analysis reviewing the application of comparable QIM or Lean and Six Sigma in all fields of surgery (*n* = 35 studies and *n* = 23) [10,11]. All papers analysed a substantial number of procedures and showed improvements in one or more process- and/or patient-related outcomes. However, the bias assessment highlighted that five of the quality improvement initiatives were rated as poor which limits the level of evidence. Nonetheless, the suitability of some of these benchmarks for quality improvement studies such as these is debatable. For example, the independent blind assessment may not be applicable for quality improvement projects, since implementing changes in clinical practice is visible to all involved.

Even though a significant number of procedures were analysed, most studies were of a single-centre study design with a pre- vs. post-intervention analysis. This may limit how representative/generalisable the outcomes are; and the consequences of applying any given QIM may vary between organisations, depending for example on the underlying problems and root causes. Nonetheless, the findings of this review indicate that multiple QIM have been effective in improving various aspects of cardiac surgical care. Therefore, it can be argued that the application of a QIM (its methodological approach) seems likely to be generalisable, even though the outcome and subsequent impact of any improvement measure(s) may vary. Indeed, one of the merits of the QIM that we have studied is their systematic approach to discovering the root cause of inefficiencies and shortcomings in a specific process, the subsequent identification and implementation of improvements based on the root cause analysis, and the long-term focus on sustaining, and even bettering the improvements.

Key areas where improvements in patient-related outcomes were seen included: ventilation time, wound infection, glycaemic control, mortality, major adverse events and discharge to home. Culig et al. reported a 61%/57% lower risk-adjusted mortality/incidence of major adverse events compared to regional rates by implementing the Toyota Production System throughout the entire care pathway [34]. Berry et al. reported improved patient outcomes [40]. Nevertheless, these results were not significant with the exception of the outcome ‘discharge location home’, which may be explained by small numbers when comparing the number of any complication (*n* = 60) to discharge location (*n* = 217). Heffner et al. reported 4.5 h shorter ventilation and Gutsche et al. reported a 3.6-h shorter median length of intubation, but without including information on, for example, actual ICU or hospital length of stay [36,41]. Meaning that while shorter ventilation times are clinically relevant, there might also be additional benefits for outcomes that were not measured. Better compliance with infection control measures, as reported by Van Tiel et al. would be more meaningful to patient care if incidence/rates of infections were also included, as in the paper of Lytsy et al. and Candis Lee Kles et al. [35,37,39].

Key process-related improvements included: increased productivity, shorter wait times, more efficient transfer of care from the operating room to the ICU and better compliance with hygiene measures. However, it is important to determine whether the reported improvements are clinically relevant to the process being studied and to patient care. For instance, shortening hand-off times by 1.9 min seems limited, but added up over the course of many procedures, it may become relevant [33].

The identified articles showed that the application of QIM can result in improved patient and process outcomes, and therefore improvements in the quality of care; increased productivity and shorter referral to treatment time help to improve access; and financial performance can be improved through higher efficiency and cost savings. A focus on improving value and quality of care for patients not infrequently leads to improved financial performance. Nonetheless, we acknowledge the narrow scope of some of these studies, the challenges in developing a high-quality evidence base for this (manufacturing-derived) subset of QIM, and the importance of organisational context in their impact and generalisability. Innovative study designs and a discussion about what constitutes high-level evidence for quality improvement, merit discussion. This is analogous to the wider discussion currently taking place about the conventional randomised controlled trial (RCT) design in medicine [43]. In addition, the investment required to implement and embed these QIM in the “organisational DNA” should not be underestimated.

To the best of our knowledge, there has not yet been a systematic review of manufacturing industry-derived QIM in cardiac surgery. However, publications concerning the application of such QIM to surgical care, in general, do exist. Nicolay et al. performed a systematic review of the application of comparable QIM in surgery [11]. Their findings are similar to our conclusion: QIM improves both patient and process-related outcomes in surgical care on different levels. In addition to the specific QIM we identified in cardiac surgery, they reported that Continuous Quality Management, Total Quality Management and Statistical Quality Control were also effective. Mason et al. conducted a systematic review regarding the application of Lean and Six Sigma in surgery. They reported that both could improve efficiency in patient flow, decrease operative complications and post-operative harms, and reduce mortality, unnecessary costs and length of stay [10]. Lee et al. evaluated recent innovations in operating room efficiency [44]. The methodologies used in this publication were Root Cause Analysis, Six Sigma and Lean. They concluded that efficiency in the surgical process can be increased through redesigning operating workflow, standardising surgical trays and teams and deploying real-time locating systems. Application of these innovations resulted in improved process-related outcomes: decreased cancellations, improved on-time starts, increased operating room utilisation, reduced turnover time and a more streamlined intra-operative process [44]. Smith et al. implemented Statistical Process Control, which is a part of Six Sigma, and reported improved OR throughput due to parallel processing [45]. Cerfolio et al. implemented Lean in a New York City Academic Hospital. They eliminated unnecessary, non-value-adding, steps through value stream mapping of the process and improved efficiency, which resulted in improved operating room turnover time [46]. These authors show that manufacturing industry-derived QIM can be successfully applied in different ways to different aspects of surgical healthcare. The success of the QIM may not be dependent on the specific QIM used, but on certain common elements within the available methods.

There are several factors for the successful implementation of QIM such as: the use of a structured problem-solving approach (e.g., PDSA), strong team engagement, a thorough understanding of the problem and goal, the use and analysis of data, and finally a culture of continuous improvement and leadership support. All projects introduced a multidisciplinary team approach to implement the QIM, which improved awareness and involvement of healthcare professionals from the complete care pathway. This has been reported as both essential and a challenge when initiating process redesign [5,47]. Engagement of stakeholders such as health care professionals leads to improved satisfaction, and thus contributes to the effectivity of implementing and sustaining change [47]. Although not explicitly evaluated by the papers in this review, it is possible that the Hawthorne effect plays a role in this [48].

Reviewing the articles presented here, identified several challenges that need to be addressed when implementing change/QIM: (1) Seven of the ten articles that were identified, applied a QIM to improve a specific part of the care pathway, for example, ventilation time or perioperative glucose control. While various independent processes can be improved separately, it may be more effective if QIM projects are applied through the complete care pathway. As an illustration, increasing productivity in the operating room alone may not be sufficient when there is inadequate capacity in the ICU to monitor patients after surgery. So, the first challenge should be to create flow through the entire care pathway. Lean and Six Sigma reduce process variation and increase predictability by standardisation. It may be argued that standardisation of processes is difficult because every patient is different; (2) therefore, the second challenge is to demonstrate that different patients can still go through standardised parts of the care pathway, creating opportunities to reduce variability. As an example, every patient in cardiac surgery requires a similar set of pre-operative diagnostics (e.g., blood samples, X-ray, transthoracic ultrasound and coronary angiography). Combining these activities in a standardised one-stop-shop process, documented in a standard operating procedure, will reduce process variation, errors, repeated work and other wastes. This results in more efficient use of resources and better patient flow. Another example of this could be the Enhanced Recovery After Surgery (ERAS) protocols, which represent an evidence-based improvement to surgical care [15]. The application of QIM to ERAS and the investigation of whether the combination may lead to further gains in the ERAS pathway represent an interesting area for further study; (3) the third challenge is to focus on improving the right metrics. A part of QIM, especially Lean and Six Sigma, is to identify meaningful metrics, which are often called key performance indicators or critical to quality characteristics. Furthermore, the applications of QIM were mostly project-based. A focus on continuous quality and process improvement needs to be embedded in organisations and is important for the sustainability of improvements, even after the project is finished. This can be monitored by continuously assessing outcomes with, for example, dashboards.

## 5. Limitations

This systematic review has several limitations. The limited and heterogeneous data precluded a quantitative analysis and the quality (level) of evidence was limited. Most studies were uncontrolled and retrospective and only half included formal statistical analysis. No study was published according to the Standards for Quality Improvement Reporting Excellence (SQUIRE) guidelines, which is recommended when publishing quality improvement projects [49]. Most of the papers were from the USA, where they have a healthcare system that is, in general, more competitive and financially driven when compared to, for example, Europe. Generalisability can be discussed for low-income countries with unstable healthcare systems. However, the goals of efficiency, value and quality are universal goals. The search strategy for this review was concerned specifically with articles focused on cardiac surgery and may not have identified more generic QIM publications that may nonetheless be relevant to cardiac surgery. There may be publication bias since unsuccessful initiatives may not have been published. Full reporting of potential conflicts of interest is important in all quality improvement studies.

## 6. Conclusions

We identified ten articles relevant to the application of recognised manufacturing industry-derived QIM to cardiac surgery. Although the studies demonstrated some magnitude of improvement in the clinical process and/or quality outcomes and therefore contributed to raising the value of care for cardiac surgery patients, the amount and quality of the evidence were limited. This supports the need for wider awareness of these QIM and for a culture of continuous quality improvement to be embraced by cardiac surgery, while at the same time highlighting that more rigorous evaluation, including how best to employ such QIM, is warranted.

## Figures and Tables

**Figure 1 jcm-11-05350-f001:**
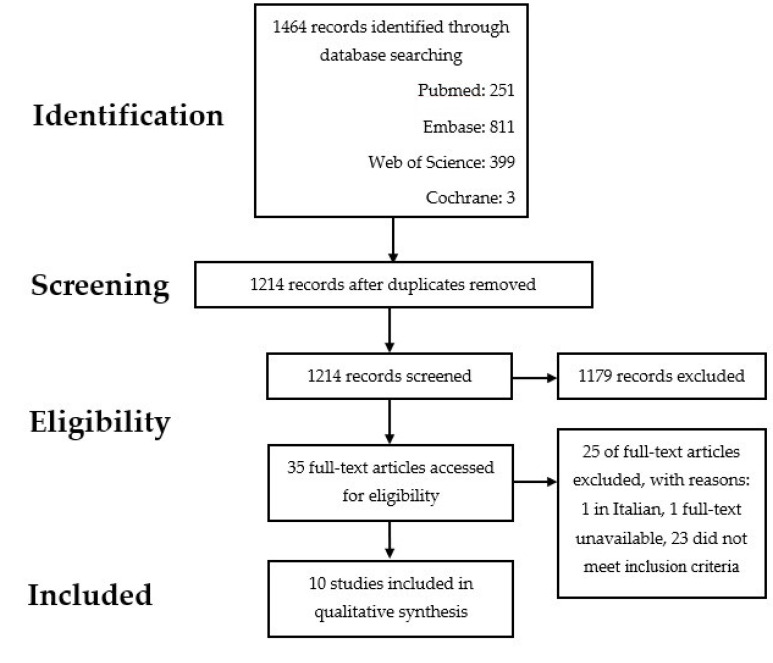
Flowchart of the search and selection procedure of studies.

**Table 1 jcm-11-05350-t001:** Results.

Author, Year, Country	Duration	Patients/Procedures	Objective	Quality Improvement Methodology	Main Interventions	Outcomes
Geoffrion et al. (2020) USA [33]	24 months (January 2015–March 2018)	Hand-offs pre-intervention (*n* = 64) and post-intervention (*n* = 62) Number of fidelity measurements (overall conformance score) (*n* = 57) Number of provider satisfaction measurements in redesign phase (*n* = 82), after 6 months (*n* = 98) and after 2.5 years (*n* = 81)	Reduce handoff (transfer of care) time	Twelve steps implementation process in four phases (planning, engaging, executing and evaluating) QIM activities: process mapping, PDSA cycles and multiple rapid-cycle process improvements	Redesign and implementation of the handoff process, implementation of handoff bundle and team training	Reduced total handoff time (in room to completion) from 12.6 ± 3.6 to 10.7 ± 2.2 min (*p* < 0.014) ‘ Improved fidelity from 18.5 ± 4.0 to 32.8 ± 9.5 (*p* < 0.001) Improved provider satisfaction after 6 months (84 vs. 80 of 100, *p* <0.02) and 2.5 years (84 vs. 87 of 100, *p* = 0.133)
Culig et al. (2011) USA [34]	28 months (March 2008–June 2010)	CABG (*n* = 253)	Improve patient outcomes, reduce costs and improve patient satisfaction	Toyota Production System QIM activities: team training, value stream mapping, pull methodology, root cause analysis, visual management, Kanban, standardisation, one-by-one processing, 5S: sort, set in order, shine, standardise and sustain, stand-up meetings	Daily stand-up meetings Collaborative bedside rounds Pre-operative briefing Intra-operative implementation of checklist, ultrasonographic aortic imaging and cerebral oximetry, handoff standardisation Post-operative protocol for medication administration, extubation and glycaemic control	Lower risk-adjusted mortality/incidence of adverse events of 61%/57% than regional rate in Society of Thoracic Surgery database Costs savings of $884,000 for CABG ($3497 per CABG)
Kles et al. (2015) USA [35]	32 months (May 2012–December 2014)	CABG (*n* = 262)	Reduce surgical site infection	Six Sigma (DMAIC) and Contextual model QIM activities: chart review, process mapping, direct observations of the process in real-time, flow-chart, standardisation, root cause analysis, contextual model	Infection prevention strategies: hair removal outside OR, routine use of mupirocin, glycaemic control, prophylactic antibiotic administration, antibiotic-impregnated sutures, soft silicone silver-impregnated dressing, dressing midsternal incision for 7 days	Reduction in incidence rate of surgical site infections from 3.74 to 0.7 per 100 procedures, and ultimately to 0 during 30 months and 590 procedures
Gutsche et al. (2014) USA [36]	12 months (July 2011–July 2012)	Cardiac surgeries total (*n* = 404), pre-intervention (*n* = 195) and post-intervention (*n* = 171)	Improve rates of early extubation	Lean methodology QIM activities: spaghetti diagram, fishbone diagram, value stream mapping, root cause analysis, PDSA	Development of extubation guideline Countermeasures: usage of air warming blankets to prevent hypothermia, use of pain scale to titrate pain medication, treatment of hypertension with antihypertensive drugs (instead of opioids), improved weaning process and availability of equipment for extubation to prevent delays	Intervention predicted extubation in <6 h improved from 27% to 50% (*p* = 0.0001) Lower median length of intubation from 9.7 to 6.1 h (*p* = 0.0019)
Lytsy et al. (2015) Sweden [37]	9 months (September 2009–July 2010)	CABG patients requiring surgical revision due to deep sternal wound infections pre-intervention (*n* = 80) and post-intervention (*n* = 13)	Illustrate that root cause analysis following by quality improvement can reduce DSWI after CABG	QIM activities: root cause analysis	Hygienic interventions in the pre-, intra- and post-operative care, e.g., hand gloves, disinfection, ultra cLean air, antibiotic prophylaxis, blood glucose control, wound dressing in place for three days	Deep sternal wound infection incidence per CABG operation decreased from 5.1% pre-intervention to 0.9% post-intervention
Watling et al. (2020) Canada [38]	24 months (September 2016–2018)	Cardiac surgery (including TAVI) pre-intervention (*n* = 788) and post-intervention (*n* = 873)	Reduce waiting times	Lean methodology QIM activities: 5-day Kaizen (rapid improvement) workshop, impact-effort analysis, weekly dashboards	Fast-tracking from ICU to ward or bypassing the ICU Improved scheduling and listing Day of surgery admission Discharge protocol	Reduced wait time with 35% from median 52 to 35 days Increased annual number of surgical interventions from 788 to 873 (10.8%) An increase in cancellations of 7.5% due to limited ICU resources
Van Tiel et al. (2006) The Netherlands [39]	Not reported (Start–Autumn 2003)	CABG OR baseline (*n* = 116), follow-up (*n* = 248) and monitoring phase (*n* = 117) Ward baseline (*n* = 16), follow-up (*n* = 22) and monitoring phase (*n* = 18)	Improve compliance with infection control measures for the care of patients during and after cardiothoracic surgery	PDSA cycles	Instruction and training of correct hygienic procedures based on infection control in the OR and on the ward Feedback on the results of baseline measurement Use of posters in the OR Presence of QI team in the OR	Overall compliance score improved in the OR and surgical ward from baseline vs. follow-up phase vs. monitoring phase
Berry et al. (2009) USA [40]	17 months (August 2005–February 2007)	CABG pre-intervention (*n* = 137) and post-intervention (*n* = 117)	To test whether process redesign by an integrated delivery system could implement evidence-based medical practices	ProvenCare programme QIM activities: multidisciplinary team meetings to review and validate best practice evidence, interview with patients, PDSA cycles	Implementation of 40 process elements (e.g., patient education materials, glycaemic control protocol, standard pre-operative anticoagulation protocol, diagnostics and medication, intra-operative time-out, documentation, antibiotic prophylaxis, and post-operative standardisation documentation, medical management, order sets)	Receiving all 40 elements in first month (59%) vs. post-intervention 100% (*p* = 0.001) Patient outcomes improved in 8 out of 9 measures (only discharge location to home significant)
Hefner et al. (2016) USA [41]	12 months (January to June 2010–January to June 2011)	CABG surgeries pre-intervention (*n* = 68) and post-intervention (*n* = 58)	Reduce prolonged mechanical ventilation after CABG surgery	Lean methodology QIM activities: gap analysis, retrospective chart review, interviews with stakeholders and focus groups, root cause analysis, standardisation	Standardised extubating protocol Dry erase boards in patients’ room to facilitate team communication Edits of post-operative ICU order set to facilitate correct medication administration	Mechanical ventilation duration reduced from 11.4 h to 6.9 h (*p* <0.001) Number of patients reintubated reduced from 11.8% to 3.5% (*p* = 0.08) Rate of prolonged ventilation decreased from 29.4% to 8.6% (*p* = 0.004)
Martinez et al. (2011) USA [42]	4 years (January 2003–March 2007)	Cardiac surgery patients admitted to CSICU total (*n* = 1892), baseline (*n* = 390) and final phase (*n* = 310) Glucose checks total (*n* = 81333), baseline (*n* = 3778) and final phase (*n* = 19043)	Generate a substantial and sustainable improvement in perioperative glucose control	Lean Six Sigma (DMAIC) QIM activities: baseline chart audit, baseline capability, process mapping, fishbone diagram, focus groups, standardisation	Perioperative insulin protocol Educational events	Admission glucose < 200 mg/dL at baseline 76% vs. final 94% (*p* <0.001) Glucose control > 6 h at baseline 0 vs. final phase 11% (*p* < 0.001) Glucose measurements increased from baseline 3 to final phase 12 per patient per day (*p* < 0.001) Hypoglycaemic events decreased from 1.7% at baseline to 0.9% at final phase (*p* < 0.001)

**Table 2 jcm-11-05350-t002:** Risk of bias assessment.

Study	Selection	Comparability	Outcome	Quality
Geoffrion et al. (2020) [33]	3	1	3	Good
Culig et al. (2011) [34]	2	1	3	Fair
Kles et al. (2015) [35]	4	0	3	Poor
Gutsche et al. (2014) [36]	4	2	3	Good
Lytsy et al. (2015) [37]	4	1	3	Good
Watling et al. (2020) [38]	4	0	3	Poor
Van Tiel et al. (2006) [39]	2	0	2	Poor
Berry et al. (2009) [40]	4	2	2	Good
Hefner et al. (2016) [41]	4	0	3	Poor
Martinez et al. (2011) [42]	4	0	3	Poor

**Table 3 jcm-11-05350-t003:** Relevant disclosures and conflicts of interests.

Study	Relevant Disclosures and Conflict of Interest
Geoffrion et al. (2020) [33]	Not mentioned
Culig et al. (2011) [34]	Supported by grant from the Highmark Foundation of Western Pennsylvania
Kles et al. (2015) [35]	None
Gutsche et al. (2014) [36]	Not mentioned
Lytsy et al. (2015) [37]	None
Watling et al. (2020) [38]	Partnership with Integrated Health Solutions, Medtronic
Van Tiel et al. (2006) [39]	Not mentioned
Berry et al. (2009) [40]	None
Hefner et al. (2016) [41]	No financial disclosures
Martinez et al. (2011) [42]	None

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
