# Peer review of "Can Quality Improvement Methodologies Derived from Manufacturing Industry Improve Care in Cardiac Surgery? A Systematic Review"

_jcm, 2022, doi:10.3390/jcm11185350_

Round 1

Reviewer 1 Report

This is an interesting paper.

The authors want to assess the usefulness of quality improvement methods for cardiosurgery. More specifically for capacity utilization and patient flow optimization.

Maybe capacity utilization and quality improvement methods are separate worlds in healthcare until now or until recently. This is different with the QIMs the authors discuss; these combine quality and efficiency methods. The authors may write something about this, as other quality methods in healthcare do not consider efficiency explicitly and systematically.

About the search strategy, in literature from the 1980’s on, often in literature the Quality Improvement methods that are referred to is generalized to ‘Japanese production systems’ without referring to the specific terms as used by the authors.

The quality of the final papers is not clear and there is a broad scope of methods studied in these articles. How representative are the studies in the articles retrieved and discussed by the authors? Probably most of these studies can be considered single case studies. Often these are useful to find useful concepts. I would suggest to pay more attention to this and use these concepts in the discussion.

The conclusion ‘QIM can improve both patient and process related outcomes in surgical care.’ , is a fairly weak conclusion. If phrased in this way it begs the question: what are the conditions that ‘can’ becomes ‘do’? I do not propose that the authors answer this question but they should be  more critical about how to study the intervention of a QIM. Some of the remarks in their discussion about success factors are not only true for the QIM methods the authors studied, but also for others. Maybe there are also Hawthorne effects?

The last part of the discussion beginning with ‘There are several challenges in healthcare that need to be addressed when implementing change ’ confuses me. I agree with the authors about this text, but what is this text? The personal view of the authors? The relation with the articles is not clear. This part is the least convincing part.

Reviewer 2 Report

Hoefsmit PC et al. presented a thorough systematic literature review focused on the potential critical role of manufacturing industry as part of quality improvement methodologies (QIM) finalized to raise the standard quality level of care in cardiac surgery.

The authors have chosen an interesting approach and have, once again, highlighted the importance of multidisciplinary approach as critical component of solving system issues. The consistent success of QIM, when observed through the light of processes not necessarily proper of the medical industry, is encouraging. The authors should be commended for making that message clear.

The manuscript could benefit from some minor proof reading and editing. In addition, the discussion may benefit from citing the role of ERAS (enhanced recovery after surgery) in cardiac surgery and how it may have contributed to improve the overall care process for patients within not just the cardiac surgery specialty, but also for surgery in general. ERAS has been subject of debate and numerous studies for the past decade and seems to be a precursor of QIM, short of manufactory industry involvement.
